# A safe and non-flammable sodium metal battery based on an ionic liquid electrolyte

Hao Sun [1], Guanzhou Zhu[1], Xintong Xu[1,2], Meng Liao[3], Yuan-Yao Li[1,4], Michael Angell[1], Meng Gu [5], Yuanmin Zhu[5,6], Wei Hsuan Hung [1,7,8], Jiachen Li[1], Yun Kuang[1,9], Yongtao Meng[1], Meng-Chang Lin[10], Huisheng Peng[3] & Hongjie Dai[1]

Rechargeable sodium metal batteries with high energy density could be important to a wide range of energy applications in modern society. The pursuit of higher energy density should ideally come with high safety, a goal difficult for electrolytes based on organic solvents. Here we report a chloroaluminate ionic liquid electrolyte comprised of aluminium chloride/1-methyl-3-ethylimidazolium chloride/sodium chloride ionic liquid spiked with two important additives, ethylaluminum dichloride and 1-ethyl-3-methylimidazolium bis(fluorosulfonyl)imide. This leads to the first chloroaluminate based ionic liquid electrolyte for rechargeable sodium metal battery. The obtained batteries reached voltages up to ~ 4 V, high Coulombic efficiency up to 99.9%, and high energy and power density of ~ 420 Wh kg$^{-1}$ and ~ 1766 W kg$^{-1}$, respectively. The batteries retained over 90% of the original capacity after 700 cycles, suggesting an effective approach to sodium metal batteries with high energy/ high power density, long cycle life and high safety.

[1] Department of Chemistry, Stanford University, Stanford, CA 94305, USA. [2] School of Aerospace Engineering, Tsinghua University, Beijing 100084, China. [3] State Key Laboratory of Molecular Engineering of Polymers, Department of Macromolecular Science, and Laboratory of Advanced Materials, Fudan University, Shanghai 200438, China. [4] Department of Chemical Engineering, National Chung Cheng University, Chia-Yi 62102, Taiwan. [5] Department of Materials Science and Engineering, Southern University of Science and Technology, Shenzhen, Guangdong 518055, China. [6] Academy for Advanced Interdisciplinary Studies, Southern University of Science and Technology, Shenzhen 518055, China. [7] Department of Materials Science and Engineering, Feng Chia University, Taichung 40724, Taiwan. [8] Institute of Materials Science and Engineering, National Central University, Taoyuan 32001, Taiwan. [9] State Key laboratory of Chemical Resource Engineering and Beijing Advanced Innovation Center for Soft Matter Science and Engineering, Beijing University of Chemical Technology, Beijing 100029, China. [10] College of Electrical Engineering and Automation, Shandong University of Science and Technology, Qingdao 266590, China. Correspondence and requests for materials should be addressed to H.D. (email: hdai@stanford.edu)

High-energy rechargeable battery systems have been actively pursued for a wide range of applications from portable electronics to grid energy storage and electric automotive industry[1–6]. At higher energies, battery safety becomes increasingly important, evident from high-profile battery fires/explosion accidents in recent years. Rechargeable batteries using flammable organic electrolytes always risk fire/explosion hazards when short circuit or thermal runaway happens, setting a bottleneck in battery design/engineering and requiring innovations of next-generation battery systems with intrinsically higher safety[7,8]. For organic electrolytes various strategies have been investigated to mitigate the safety concerns, including the use of voltage or temperature-sensitive separators[9–11] and overcharge protection additives[12]. Developing new electrolyte systems that are intrinsically non-flammable has also been actively pursued[13–15]. In particular, room temperature ionic liquids (ILs) have been widely explored as promising candidates due to their non-flammable nature[16–18]. Among them, ILs comprised of $AlCl_3$ and 1-ethyl-3-methylimidazolium chloride ([EMIm]Cl) are a classical chloroaluminate based electrolyte system with many desired properties including non-flammability, non-volatility, low viscosity, high conductivity, and high thermal stability and chemical inertness[17,19]. In this electrolyte, $AlCl_3$ complexes with the Cl ion from [EMIm]Cl to produce $AlCl_4^-$ and $EMIm^+$, and any excess $AlCl_3$ converts a portion of $AlCl_4^-$ into $Al_2Cl_7^-$, resulting in the coexistence of $AlCl_4^-$ and $Al_2Cl_7^-$:

$$AlCl_3 + [EMIm]Cl \rightarrow AlCl_4^- + [EMIm]^+ \quad (1)$$

$$AlCl_4^- + AlCl_3 \rightarrow Al_2Cl_7^- \quad (2)$$

The $AlCl_3$/[EMIm]Cl-based ILs have been used as electrolytes for rechargeable metal batteries[20,21]. An example was rechargeable aluminum-graphite battery developed by our group and others with fast and highly reversible $AlCl_4^-$ intercalation/de-intercalation into graphite positive electrode, and $Al_2Cl_7^-$ plating and stripping on Al negative electrode[20]. Nevertheless, it is desirable to develop higher voltage and higher energy density battery systems utilizing chloroaluminate IL electrolytes. A promising strategy is replacing Al by more reactive metal negative electrodes with lower standard electrode potentials such as sodium and lithium, which could raise the battery voltage and allow the use of well-established positive electrode materials with higher energy densities. Indeed, researchers have been pursuing this direction since almost 30 years ago. In as early as 1990, Melton et al. reported the first buffered $AlCl_3$/[EMIm]Cl IL system by adding NaCl, eliminating $Al_2Cl_7^-$ and introducing Na ions into the electrolyte[22] via

$$Al_2Cl_7^- + NaCl \rightarrow 2AlCl_4^- + Na^+ \quad (3)$$

Thus far, however, reversible and stable deposition and stripping/oxidation of Na metal in buffered $AlCl_3$/[EMIm]Cl ILs towards rechargeable Na batteries have been hindered, with or without the use of a variety of electrolyte additives such as HCl[23], [EMIm]HCl2[24,25], triethanolamine hydrochloride[26] and thionyl chloride[27]. These additives can stabilize Na redox to limited degrees, affording Coulombic efficiencies (CEs) of 65–94% for Na plating/stripping[23–27]. For instance, the CE record of reversible Na redox was 94% achieved with ~ 6 Torr HCl added to NaCl-buffered $AlCl_3$/[EMIm]Cl = ~ 1.7 IL at 6.4 mA cm$^{-2}$, but it rapidly decayed at a lower current density[23]. None of the chloroaluminate ILs could afford multicycle Na plating/stripping with sufficiently high CE to pair with sodium positive electrode for Na battery cells[17].

Here we present an ionic liquid electrolyte based on NaCl-buffered $AlCl_3$/[EMIm]Cl for safe and high energy Na batteries. Two electrolyte additives at the 1 to 4% by mass

level i.e., ethylaluminum dichloride ($EtAlCl_2$) and 1-ethyl-3-methylimidazolium bis(fluorosulfonyl) imide ([EMIm]FSI) are key to stabilizing SEI on sodium negative electrode for reversible Na plating/stripping. In a Na/Pt cell containing this IL electrolyte, a CE of ~95% is reached at 0.5 mA cm$^{-2}$ over ~ 100 reversible Na plating/stripping cycles. With the optimized IL electrolyte, we pair Na negative electrode with sodium vanadium phosphate (NVP) and sodium vanadium phosphate fluoride (NVPF) based positive electrodes to afford high discharge voltage up to ~4 V, high CEs up to 99.9 %, and maximal energy and power density of 420 Wh kg$^{-1}$ and 1766 W kg$^{-1}$, respectively based on active material mass of positive electrode. In addition, more than 90% of the original capacity is retained after over 700 cycles. Solid-electrolyte interphase (SEI) analysis reveals SEI compositions including NaCl, $Al_2O_3$ and NaF derived from the reactions between Na and the anions in the IL electrolyte. The results shed light on future electrolyte and SEI design towards practical sodium metal batteries with high safety and high energy/power densities.

## Results

**Properties of NaCl-buffered $AlCl_3$/[EMIm]Cl ionic liquid.** Preparation of IL electrolyte (see "Methods" section) started by mixing anhydrous $AlCl_3$ and [EMIm]Cl at a molar ratio of 1.5:1 to form an acidic room-temperature IL ($AlCl_3$/[EMIm]Cl = 1.5), followed by buffering to neutral with excess NaCl and then adding 1 wt% $EtAlCl_2$ and 4 wt% [EMIm]FSI to afford the final NaCl-buffered chloroaluminate IL electrolyte (referred as 'buffered Na–Cl–IL electrolyte') (Fig. 1a). Raman spectroscopy was performed to probe the evolution of $AlCl_4^-$ and $Al_2Cl_7^-$ species in the IL at different stages (Fig. 1b). Both $AlCl_4^-$ and $Al_2Cl_7^-$ peaks were observed in the starting acidic IL with $AlCl_3$/[EMIm]Cl = 1.5. After NaCl buffering of the electrolyte to neutral, the $Al_2Cl_7^-$ peaks at 309 and 430 cm$^{-1}$ disappeared while the $AlCl_4^-$ peak at 350 cm$^{-1}$ strengthened, indicating the conversion of $Al_2Cl_7^-$ to $AlCl_4^-$ by NaCl on the basis of equation (3). Subsequent addition of 1 wt% $EtAlCl_2$ resulted in a noticeable further enhancement of the $AlCl_4^-$ peak. This was attributed to reactions of $EtAlCl_2$ with trace amounts of protons and undissolved NaCl in the buffered $AlCl_3$/[EMIm]Cl = 1.5 IL with the generation of $AlCl_4^-$, $C_2H_6$ and $Na^+$ via [28]:

$$EtAlCl_2 + H^+ + 2NaCl \rightarrow C_2H_6(g) + AlCl_4^- + 2Na^+ \quad (4)$$

No obvious change in the Raman spectrum of chloroaluminate species was observed after the addition of 4 wt% [EMIm]FSI (Fig. 1b). The final buffered electrolyte (named buffered Na–Cl–IL hereon) was comprised of $Na^+$, $AlCl_4^-$, $EMIm^+$ and $FSI^-$ with $Na^+$ and $FSI^-$ molar concentration of ~ 1.76 M and ~0.2 M, respectively.

An important property of the buffered Na–Cl–IL was its high ionic conductivity of ~9.2 mS cm$^{-1}$ at 25 °C, which was 2–20 times higher than those of previously reported IL electrolytes based on bulky cations (e.g., N-butyl-N-methylpyrrolidinium and N-propyl-N-methylpyrrolidinium) for Na batteries[29–32] (Fig. 1c). The ionic conductivity was comparable to conventional organic electrolytes, for example, ~6.5 mS cm$^{-1}$ of 1 M $NaClO_4$ in propylene carbonate (PC), and 6.35 mS cm$^{-1}$ of 1 M $NaClO_4$ in ethylene carbonate/diethyl carbonate (EC/DEC, 1:1 by weight)[33]. The thermal stability of our buffered Na–Cl–IL electrolyte was compared with a conventional organic electrolyte 1 M $NaClO_4$ in EC/DEC (1:1 by vol) with 5 wt% FEC additive by thermogravimetric analysis (TGA) (Fig. 1d). The organic electrolyte showed a rapid weight loss above 132 °C, and lost ~85% of the original weight at 230 °C due to decomposition of the carbonate solvents

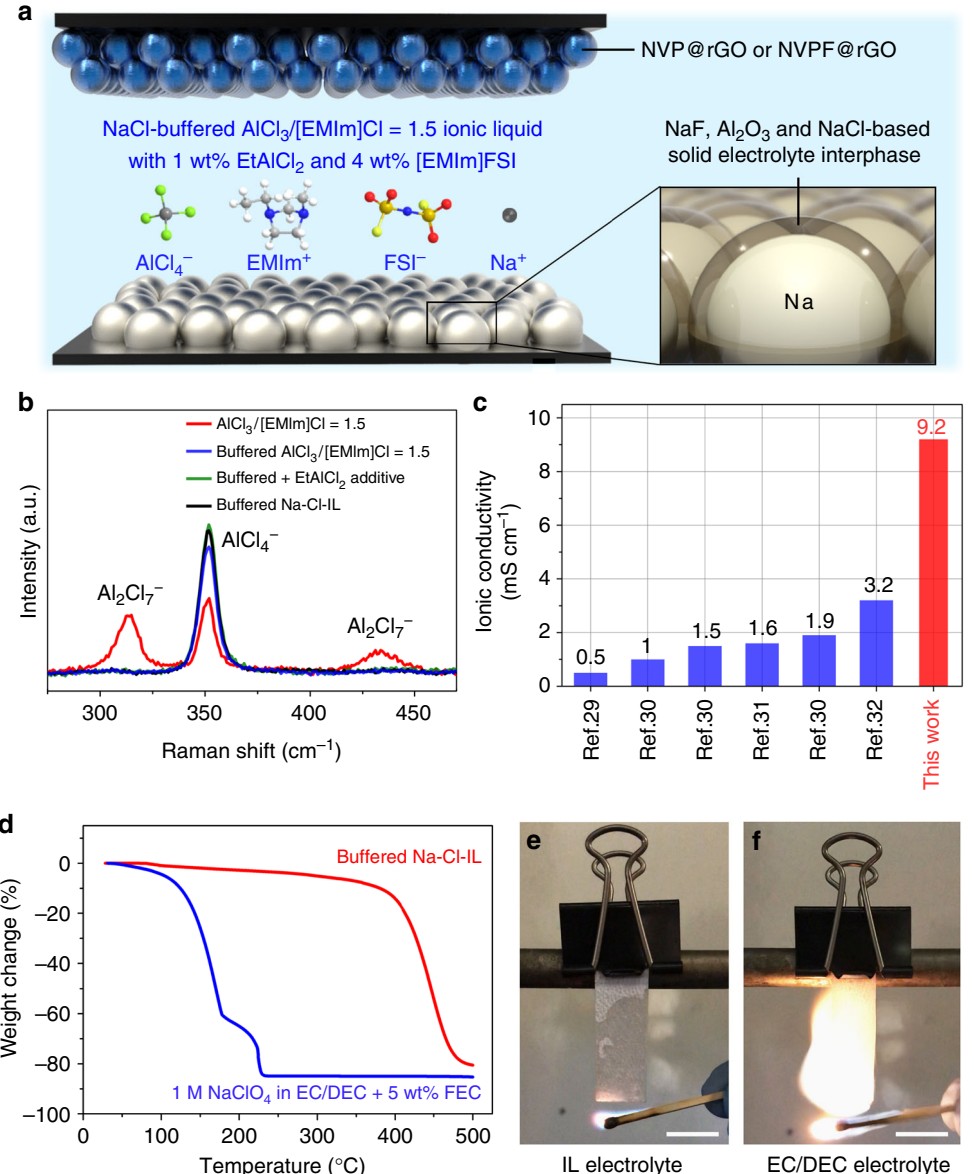

**Fig. 1** Properties of the buffered Na–Cl–IL electrolyte. **a** Schematic illustration of the battery configuration and electrolyte composition of the IL electrolyte. **b** Raman spectra of ILs based on $AlCl_3/[EMIm]Cl = 1.5$ with different additives. **c** Ionic conductivities of buffered Na–Cl–IL and other IL-based electrolytes for Na batteries at 25 °C[29–32]. **d–f** Thermal stability (**d**) and flammability tests using buffered Na–Cl–IL (**e**) and conventional 1.0 M $NaClO_4$ in EC:DEC (1:1 by vol) with 5 wt% FEC electrolytes (**f**). Scale bars in (**e**, **f**), 1 cm

in this temperature range. In comparison, the buffered Na–Cl–IL showed a much better thermal stability without severe weight loss until ~400 °C. The non-flammable nature of the buffered Na–Cl–IL electrolyte was confirmed when it was soaked into a porous separator and contacted with flame (Fig. 1e and Supplementary Movie 1) without causing fire. In contrast, the organic carbonate electrolyte easily caught fire and burned immediately (Fig. 1f and Supplementary Movie 2).

**Electrochemistry of Na–Cl–IL electrolyte**. In a Na vs. carbon-fibre-paper cell containing the buffered Na–Cl–IL electrolyte, linear sweep voltammetry scan was performed (Fig. 2a) and revealed a pair of sodium redox peaks on the cathodic side and no obvious electrolyte decomposition was observed until ~ 4.56 V on the anodic side, indicating high electrochemical stability of the electrolyte for high-voltage sodium battery systems. Sodium reduction/oxidation peaks were clearly observed in cyclic

voltammetry (CV) with a Pt working electrode, a Na reference and counter electrode in buffered Na–Cl–IL electrolyte, showing reversible Na plating and stripping on Pt (Fig. 2b). In striking contrast, redox peaks were completely missing in buffered electrolyte without [EMIm]FSI additive, suggesting its critical role of stabilizing Na plating/stripping (Fig. 2c). Galvanostatic charge-discharge test investigated Na plating/stripping on Pt in buffered Na–Cl–IL electrolyte at a plating current density of 0.5 mA cm$^{-2}$ for 30 min. The CE increased from ~72 to ~91% during the first 5 cycles for SEI formation and then reached ~95%, which is a new record of Na redox for both buffered chloroaluminate ILs and any other ionic liquids based on different cations (including benzyldimethylethylammonium, butyldimethylpropylammonium, trimethylhexylammonium, dibutyldimethylammonium and *N*-butyl-*N*-methylpyrrolidinium) and anions (including FSI and TFSI)[17,34–36] (Fig. 2d, TFSI represents bis(trifluoromethanesulfonyl)imide). Reversible Na plating/stripping

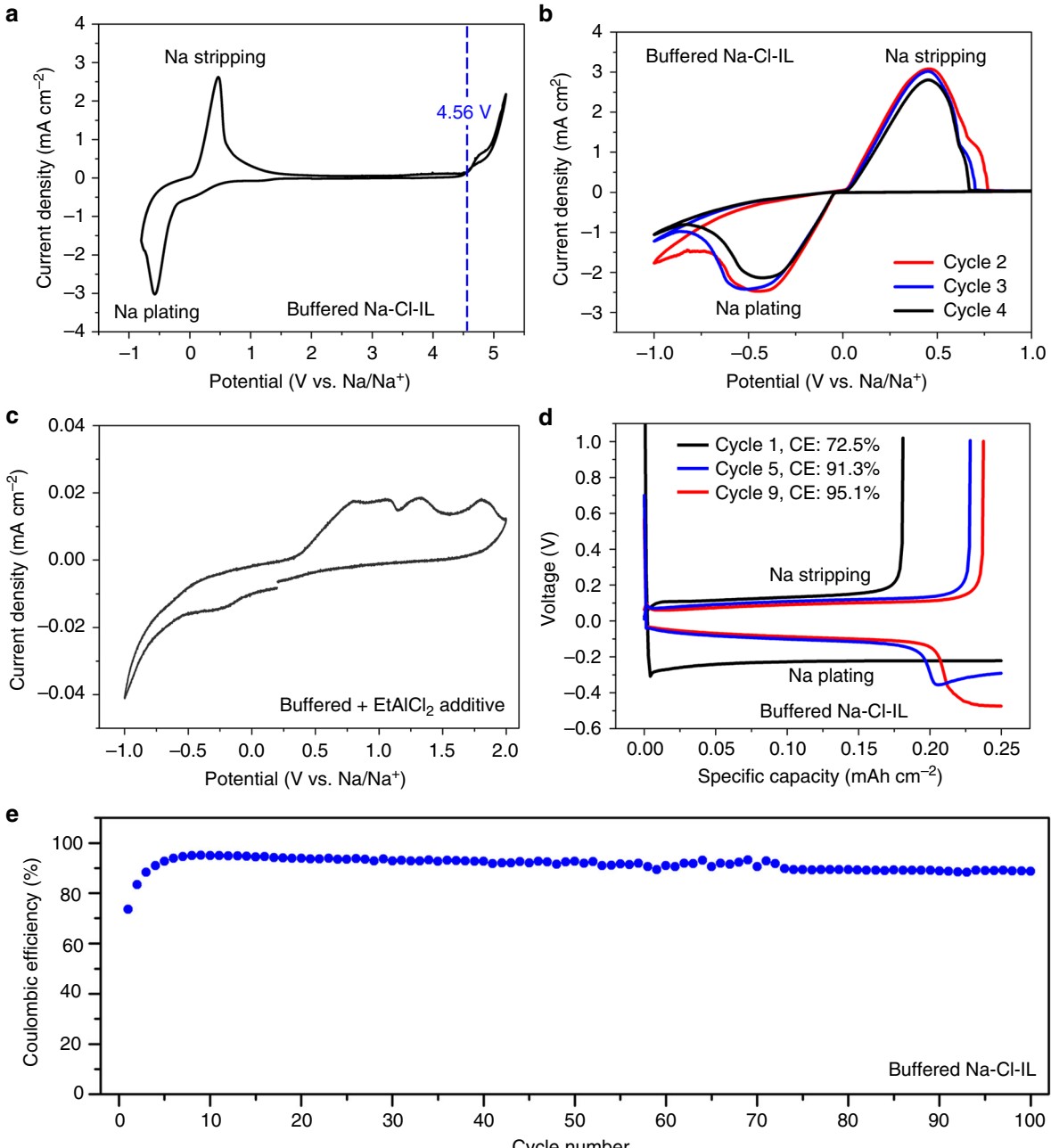

**Fig. 2** Electrochemical properties of the buffered Na–Cl–IL electrolyte. **a** Linear sweep voltammetry profile of buffered Na–Cl–IL electrolyte. Working electrode, carbon fibre paper. Counter and reference electrode, Na foil. Scan rate, 2 mV s$^{-1}$. **b**, **c** CV curves of Na/Pt cells using buffered + EtAlCl$_2$ additive and buffered Na–Cl–IL electrolyte at a scan rate of 2 mV s$^{-1}$, respectively. **d** Na plating/stripping profiles of Na/Pt cells using buffered Na–Cl–IL electrolyte at a current density of 0.5 mA cm$^{-2}$. **e** Na plating/stripping Coulombic efficiency of a Na/Pt cell using Buffered Na–Cl–IL electrolyte at 0.5 mA cm$^{-2}$. The plating capacity in (**d**, **e**): 0.25 mAh cm$^{-2}$

cycling was performed for 100 cycles (Fig. 2e), which was the first-time multicycle Na redox was performed in buffered AlCl$_3$/[EMIm]Cl ILs. Without [EMIm]FSI additive in buffered AlCl$_3$/[EMIm]Cl = 1.5 electrolyte, plating current was observed but without observable stripping capacity (Supplementary Fig. 1).

The morphology of the plated Na on Cu after five plating/stripping cycles at a current density of 0.5 and 1.5 mA cm$^{-2}$ was investigated by scanning electron microscopy (SEM), showing particle sizes ranging from 5 to 10 μm without obvious dendritic morphology (Supplementary Fig. 2). The inner part of the Na particle was analyzed using focused ion beam (FIB) to expose cross-section of the interior (Supplementary Fig. 3). EDS element

mapping of the cross-section revealed the existence of Na as the major element, together with O, Al, F and C, and very little Cl was detected inside the particle, suggesting the distribution of Cl mainly on the surface of Na rather than inside (Supplementary Fig. 4). More detailed analysis of SEI on sodium negative electrodes are shown later in this paper.

**Na metal cells based on Na–Cl–IL electrolyte.** Next, we made a Na metal battery by pairing a Na negative electrode with a positive electrode formed by coating Na$_3$V$_2$(PO$_4$)$_3$@reduced graphene oxide (NVP@rGO) particles on a carbon-fibre-paper

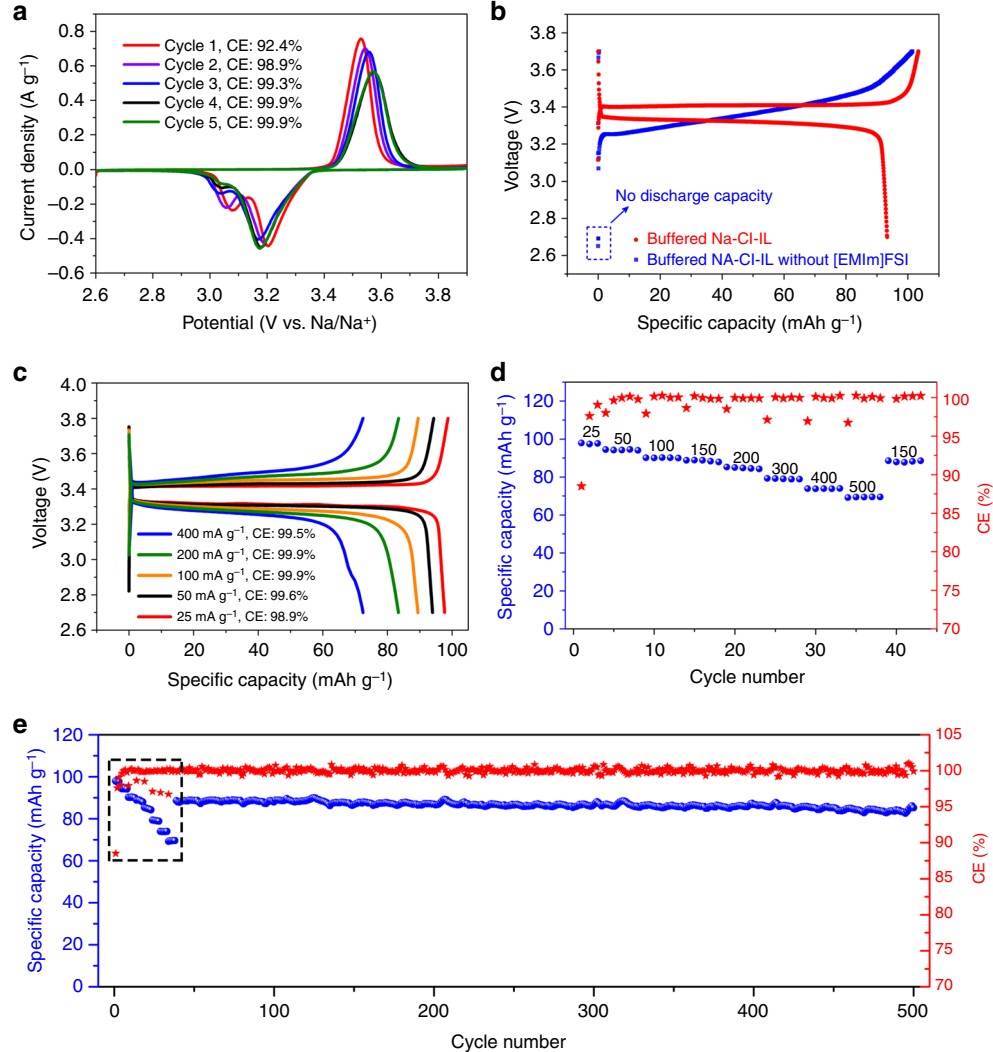

**Fig. 3** Na/NVP/@GO cell performances in buffered Na–Cl–IL electrolyte. **a** CV curves of a Na/NVP@rGO cell using buffered Na–Cl–IL electrolyte at a scan rate of 2 mV s$^{-1}$. **b** Initial galvanostatic charge-discharge curves of a Na/NVP@rGO cell using buffered Na–Cl–IL electrolytes with and without [EMIm]FSI additive at 25 mA g$^{-1}$. **c** Galvanostatic charge-discharge curves of a Na/NVP@rGO cell using buffered Na–Cl–IL electrolyte at varied current densities from 25 to 400 mA g$^{-1}$. **d**, **e** Rate and cyclic stability of a Na/NVP@rGO cell using buffered Na–Cl–IL electrolyte. The boxed region of (**e**) corresponds to the rate performance of (**d**) at varied current densities from 20 to 500 mA g$^{-1}$. After that, a current density of 150 mA g$^{-1}$ was used for cycling

substrate (see Method). NVP was a widely explored positive electrode material for rapid and reversible Na ion insertion/de-insertion in its lattice, and the interconnected conducting network formed by rGO sheets further enhanced the charge transfer process[37,38]. Powder X-ray diffraction (XRD) measurements showed a NASICON-type framework with R$\bar{3}$c space group with high crystallinity of the synthesized NVP@rGO particles (Supplementary Fig. 5). SEM and transmission electron microscopy (TEM) showed NVP particles several hundred micrometers in size blended with rGO sheets (Supplementary Figs. 6 and 7). The lattice fringes with *d*-spacings of 0.44 and 0.34 nm were assigned to the (104) planes of rhombohedral NVP and (002) planes of multi-layered rGO respectively[37]. The rGO content of the NVP@rGO hybrid was around 1.1 wt% determined by thermogravimetric analysis (TGA, Supplementary Fig. 8).

Cyclic voltammetry of a Na/NVP@rGO cell with the optimized buffered Na–Cl–IL electrolyte (see supplementary Fig. 9 for electrolyte optimization) showed a pair of oxidation and reduction peaks corresponding to the redox reactions of V$^{3+}$/V$^{4+}$ couples, and the CE increased to ~99.9 % within four

cycles and then stabilized (Fig. 3a). A mass loading of NVP@rGO ~ 3.0 mg cm$^{-2}$ was used unless specified otherwise. A charge-discharge plateau at ~ 3.4 V was seen with a specific discharge capacity of 93.3 mA g$^{-1}$ based on the mass of NVP@rGO at a rate of 25 mA g$^{-1}$ (Fig. 3b). In striking contrast, the buffered Na–Cl–IL electrolyte without [EMIm]FSI additive showed a negligible discharge capacity (0.03 mAh g$^{-1}$) (Fig. 3b). The Na/NVP@rGO cell in buffered Na–Cl–IL electrolyte showed good rate capabilities at higher rates (Fig. 3c), with a specific discharge capacity of ~ 70 mAh g$^{-1}$ at 500 mA g$^{-1}$ (~4.3 C), which was ~ 71% of the specific capacity at 25 mA g$^{-1}$ (Fig. 3d). The Na/NVP@rGO cell could retain ~96 % of the initial capacity for over 460 cycles at 150 mA g$^{-1}$ (~ 0.4 mA cm$^{-2}$) with a high average CE of 99.9 % (Fig. 3e). This was the first time > 99 % CE was achieved for Na metal battery in buffered chloroaluminate IL electrolytes. In comparison, a Na/NVP@rGO cell based on a conventional organic carbonate electrolyte, 1 M NaClO$_4$ in ethylene carbonate/diethyl carbonate (EC/DEC, 1:1 by vol) with 5 wt% fluoroethylene carbonate (FEC) only retained 79 % of the initial capacity after 450 cycles at 150 mA g$^{-1}$ (Supplementary

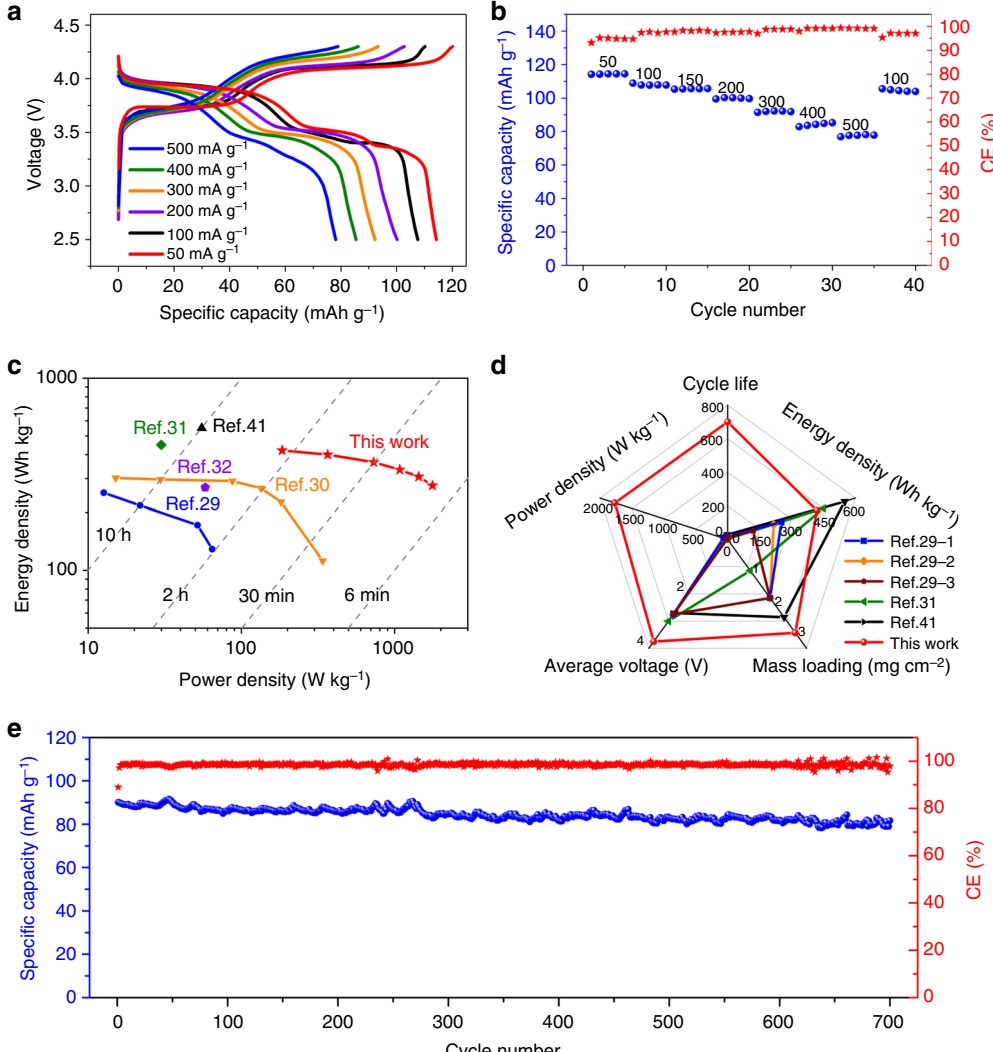

**Fig. 4** Na/NVPF@GO cell performances in buffered Na–Cl–IL electrolyte. **a** Galvanostatic charge-discharge curves of a Na/NVPF@rGO cell at varied current density from 50 to 500 mA g$^{-1}$. **b** Capacity and Coulombic efficiency retention of a Na/NVPF@rGO cell when cycled at different current densities from 50 to 500 mA g$^{-1}$. **c**, **d** Ragone and Radar plots of this work compared with other reported room-temperature Na batteries based on IL electrolytes, respectively[29-32,41]. The specific capacity, energy and power density in this work and previous literatures were all calculated based on the mass of active materials on positive electrode. The cycle life in (**d**) is determined by the cycle number when the capacity dropped below 90% of the original capacity[29]. 1, 2 and 3 represent three different IL electrolytes based on 1 M NaBF$_4$, NaClO$_4$ and NaPF$_6$ salts, respectively. **e** Cyclic stability of a Na/NVPF@rGO cell using buffered Na–Cl–IL electrolyte at 300 mA g$^{-1}$

Fig. 10), which is significantly lower than ∼96 % based on buffered Na–Cl–IL electrolyte under the same condition. A similarly high average CE of 99.9 % was demonstrated in organic electrolyte when the cell was stably cycled, but CE fluctuation was observed after 400 cycles (Supplementary Fig. 10). The Na/NVP@rGO cell based on buffered Na–Cl–IL electrolyte realized an approximate 100-cycle longer cycle life compared with that using conventional organic electrolyte. With an increased NVP@rGO mass loading of 8.0 mg cm$^{-2}$, a specific discharge capacity of ∼92 mAh g$^{-1}$ was delivered at 25 mA g$^{-1}$ using buffered Na–Cl–IL electrolyte, corresponding to 94 % of the capacity with 3.0 mg cm$^{-2}$ loading (Supplementary Fig. 11). A slightly lower CE of ∼99.0% was demonstrated at the loading of 8.0 mg cm$^{-2}$ compared with ∼99.9% at 3.0 mg cm$^{-2}$.

With a stable voltage window up to ∼4.6 V (Fig. 2a), the buffered Na–Cl–IL electrolyte was compatible with higher voltage positive electrodes such as Na$_3$V$_2$(PO$_4$)$_2$F$_3$@rGO to afford Na metal battery cells with higher discharge voltage and energy density. We synthesized NVPF@rGO by a facile hydrothermal

method, which was the first time NVPF@rGO hybrid was prepared via a one-step and low-temperature (120 °C) method without any freeze drying or annealing treatments[39] (see Method). XRD patterns (Supplementary Fig. 12) indicated the prepared NVPF and NVPF@rGO mainly comprised of tetragonal Na$_3$V$_2$(PO$_4$)$_2$F$_3$ (ICDD PDF No. 01-089-8485)[40] with an average size of ∼100 nm. The NVPF particles were uniformly hybridized with rGO sheets, affording an interconnected conducting network to enhance electron transfer (Supplementary Fig. 13). The rGO content of the NVPF@rGO hybrid was around 4.4 % verified by TGA (Supplementary Fig. 14). Two pairs of oxidation and reduction peaks (3.75 V/3.5 V and 4.12 V/3.91 V) were observed in the CV curves of the positive electrode, corresponding to redox reactions of V$^{3+}$/V$^{4+}$ and couples respectively (Supplementary Fig. 15). Compared to NVP@rGO with V$^{3+}$/V$^{4+}$ redox, the introduction of fluorine in NVPF@rGO allowed stable V$^{4+}$/V$^{5+}$ redox[40], affording a higher charge/discharge plateau at ∼4 V. The Na/NVPF@rGO cell based on buffered Na–Cl–IL electrolyte demonstrated good rate performances under 50 to 500 mA g$^{-1}$

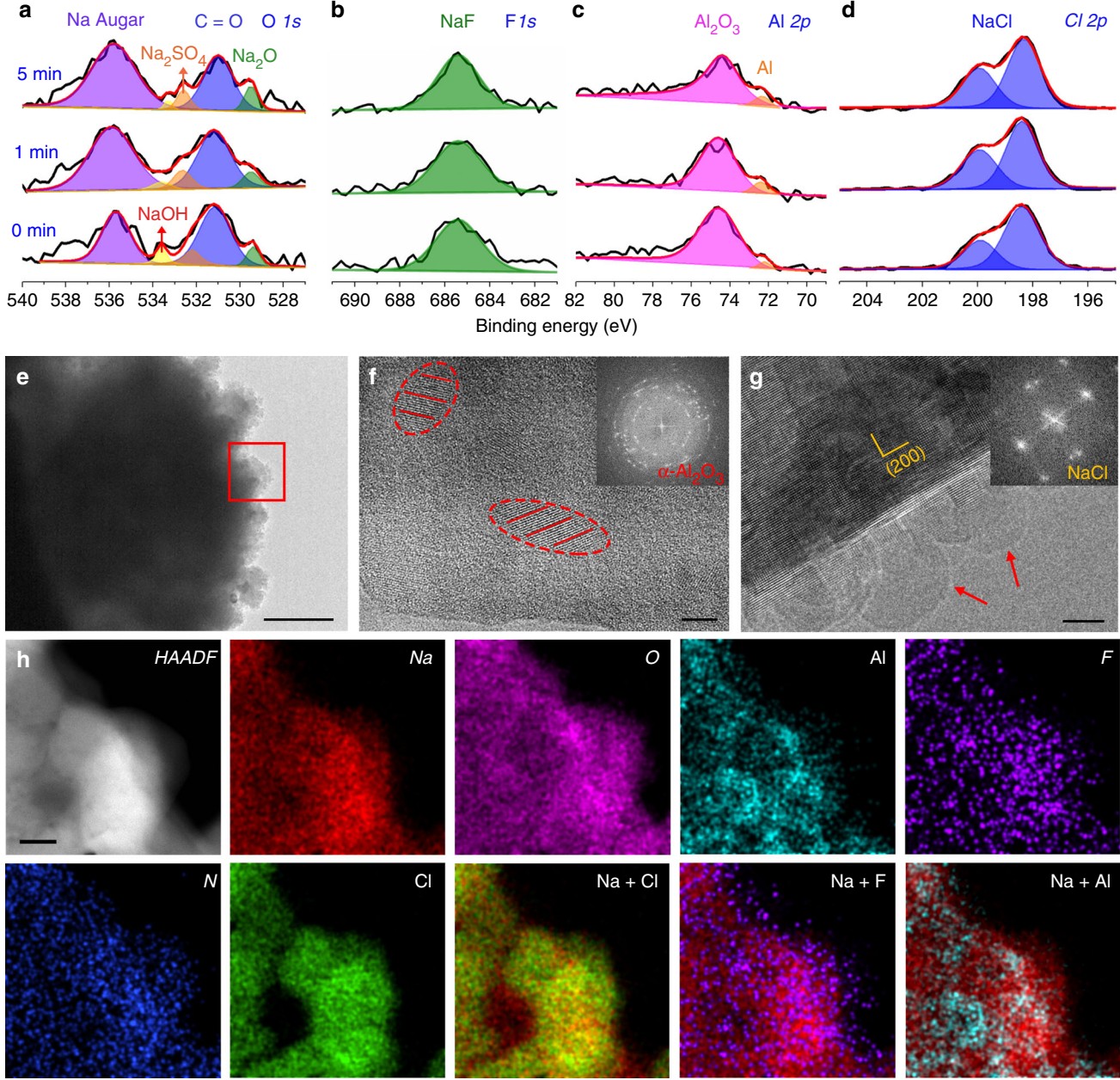

**Fig. 5** Morphology and solid-electrolyte interphase (SEI) probing of the plated Na in buffered Na–Cl–IL electrolyte. **a–d** High-resolution XPS spectra for Na Auger and O1s (**a**), F 1s (**b**), Al 2p (**c**) and Cl 2p (**d**) of the Na negative electrode from a Na/NVP@rGO cell with NVP@rGO mass loading of 5.0 mg cm$^{-2}$ at different depths, respectively. The cell was cycled at 100 mA g$^{-1}$ (~0.5 mA cm$^{-2}$) for 20 cycles and stopped at fully charged state prior to characterization. **e** Cryo-TEM image of Na-plated Cu grid at a current density of 0.1 mA cm$^{-2}$. Scale bar, 500 nm. **f, g** High-resolution Cryo-TEM images and diffraction patterns (inset) of SEI concerning Al$_2$O$_3$ and NaCl. Scale bars in (**f, g**), 5 nm. **h** High-angle annular dark-field (HAADF) and the corresponding element mapping images for SEI composition probing using STEM. Scale bar, 100 nm

(0.16 to 1.6 mA cm$^{-2}$) current densities and CEs from 95 % to ~ 99 % (Fig. 4a, b). The maximal energy density was ~ 420 Wh kg$^{-1}$ based on the mass of NVPF@rGO. With an increase of NVPF mass loading from 3.0 to 8.0 mg cm$^{-2}$, both the specific capacity and energy density were well retained, with an energy density of ~ 394 Wh kg$^{-1}$ at a mass loading of NVPF@rGO ~8.0 mg cm$^{-2}$ operated under a 50 mA g$^{-1}$ (~0.4 mA cm$^{-2}$) current (Supplementary Fig. 16). The NVPF@rGO positive electrode showed high energy density at various rates in the buffered Na–Cl–IL electrolyte (Fig. 4c), delivering an energy density of 276 Wh kg$^{-1}$ in ~ 10 min discharging time, corresponding to a power density of 1766 W kg$^{-1}$ based on the mass of NVPF@rGO at a current density of 500 mA g$^{-1}$ (~ 1.6 mA cm$^{-2}$). The superior rate

performance over previous NVPF-based positive electrodes in IL electrolytes[29–32,41] was attributed to the 2- to 20-fold higher ionic conductivity of the Na–Cl–IL electrolyte, and the novel NVPF@rGO hybrid that facilitated charge transfer[42,43].

The Na/NVPF@rGO cell with a NVPF@rGO mass loading of 3.0 mg cm$^{-2}$ showed excellent cycling stability in our IL electrolyte, retaining more than 90% of the initial specific capacity over 710 cycles at a current density of 300 mA g$^{-1}$ (~ 0.81 mA cm$^{-2}$) with an average CE of 98.5% (Fig. 4e). At a higher NVPF@rGO mass loading of 5.3 mg cm$^{-2}$, a Na/NVPF@rGO cell could retain 91% of the initial specific capacity after 360 galvanostatic charge-discharge cycles at 150 mA g$^{-1}$ (~0.7 mA cm$^{-2}$) with an average CE of 98.2% (Supplementary

Fig. 17). The key performance parameters of the Na/NVPF@rGO cell in buffered Na–Cl–IL electrolyte, including energy/power density, cycle life, discharge voltage and mass loading outperformed previous cells based on room-temperature IL electrolytes[29,31,41] (Fig. 4d and Supplementary Table 1).

The EtAlCl$_2$ additive was found important to enhance the cycling stability of Na batteries with Na–Cl–IL electrolyte, when comparing two Na/NVPF@rGO cells in IL electrolytes with and without 1 wt% EtAlCl$_2$ (Supplementary Fig. 18). The presence of EtAlCl$_2$ additive improved cycle life by ~500 cycles at 300 mA g$^{-1}$, which could be explained by the elimination of trace amounts of residual protons and free chloride ions in the electrolyte via equation (4).

**Solid-electrolyte interphase chemistry of Na–Cl–IL electrolyte.** It is well known that SEI plays a critical role in stabilizing the interface between alkali metal negative electrodes and electrolytes[44–46]. Due to the unusual composition of our IL electrolyte, the SEI chemistry could be different from that in conventional organic electrolytes. To this end, we first analyzed the elemental composition and depth profile by X-ray photoelectron spectroscopy (XPS) of a Na negative electrode from a Na/NVP@rGO cell with the mass loading of NVP@rGO 5.0 mg cm$^{-2}$. The cell was cycled for 20 cycles at 100 mA g$^{-1}$ (~0.5 mA cm$^{-2}$) and stopped at a fully charged state (Na plated on negative electrode). Surface XPS profile identified the presence of Na, O, C, Cl, F, Al and N (Supplementary Fig. 19). XPS profiling by Ar sputtering showed pronounced Na Auger peak at 535.7 eV at all sample depths (Fig. 5a). The O 1s peaks at 531.2, 529.4, 532.2 and 533.6 eV indicated the presence of Na$_2$CO$_3$, Na$_2$O, Na$_2$SO$_4$ and NaOH, respectively (Fig. 5a). The presence of NaOH was only at the surface, as it was generated from the contamination by water when the sample was briefly exposed to air during transfer to XPS. Part of the Na$_2$CO$_3$ could also be from reaction with water and carbon dioxide in air and decreased in intensity after sputtering. In contrast, the intensity of Na$_2$O and Na$_2$SO$_4$, formed by FSI anion and sodium metal showed no obvious decrease during sputtering, indicating their existence in SEI. As expected, the F 1s peak at ~685.5 eV confirmed the presence of NaF as the major F-based SEI (Fig. 5b). The FSI anions in [EMIm]FSI were responsible for F-based SEI via reactions with the highly reactive Na metal, consistent with previous literature[47,48]. The Al 2p peaks at 74.5 eV indicated the presence of Al$_2$O$_3$ as a major Al-based SEI component with a small portion of metallic Al observed (Fig. 5c). The two pronounced peaks at ~198.4 and 199.8 eV corresponded to Cl 2p$_{1/2}$ and Cl 2p$_{3/2}$ peaks, suggesting NaCl as another major SEI component (Fig. 5d). The weak N 1s peak at ~400 eV indicated the presence of N-based species generated from the decomposition of FSI anion (Supplementary Fig. 20), consistent with previous literature based on LiFSI-based organic electrolytes[49,50]. Overall, a hybrid SEI formed on sodium metal comprised of NaF, Na$_2$O, Na$_2$SO$_4$, Al$_2$O$_3$, Al and NaCl contributed to the reversible plating/stripping process of Na in buffered Na–Cl–IL electrolyte.

To gain a deeper insight into the Na plating process in buffered Na–Cl–IL electrolyte, cryogenic transmission electron microscope (Cryo-TEM) was used to probe the morphology and elemental composition of plated Na on Cu grids without exposing the sample to air (see "Methods" section). Cryo-TEM was demonstrated recently as a powerful tool in probing the morphological and component information of beam-sensitive battery materials such as Li metal[51,52], but not yet used for investigating SEI on sodium thus far. We first investigated the initial Na plating on a Cu grid, which involved Na growth and SEI formation at the initial stage. The plated Na (without exposing to air)

demonstrated a spherical morphology (Fig. 5e). High-resolution image showed some clusters in SEI with clear lattice fringes showing a $d$-spacing of 0.347 nm indexed to the (012) planes of α-Al$_2$O$_3$, which was also confirmed by diffraction pattern (Fig. 5f). In addition, the compact stacking of many nanocubes with an average size of ~10 nm was observed on the edge of SEI, with lattice fringes at a $d$-spacing of 0.284 nm indexed to (200) planes of NaCl and corroborated by diffraction pattern (Fig. 5g).

Element mapping analysis on these regions was performed using scanning transmission electron microscopy (STEM), indicating the presence of Na, O, Cl, Al, F and N that was in accordance with the XPS results, confirming the hybrid SEI composition of this novel IL electrolyte (Fig. 5h). The overlapped Na and Cl mapping indicated the presence of NaCl, which was consistent with the stacking cubes and diffraction pattern of NaCl detected in Cryo-TEM (Fig. 5f). The F mapping mainly distributed in the region near the surface, and showed a good overlap with Na mapping, which was in accordance with the XPS results that indicated the presence of NaF layer. The merged Na and Al mapping showed the aggregation of Al with the formation of some Al clusters, rather than distribute uniformly with Na in the SEI matrix (Fig. 5h). It can be explained by the fact that Al and Na cannot form an alloy; thus, Al might prefer to plate on Al rather than Na, which could account for the interconnected structure of Al observed in the mapping image.

**Discussion**
Compared with previous IL electrolytes for Na cells, the Na–Cl–IL electrolyte system is interesting in several ways. First, the high ionic conductivity (~9.2 mS cm$^{-1}$ at 25 °C) outperforms previously reported IL electrolytes based on bulky cations (e.g., benzyldimethylethylammonium and N-butyl-N-methylpyrrolidinium) and anions (e.g., FSI$^-$ and TFSI$^-$), allowing for both high-energy density and rate capability/power density of the Na metal cells (Supplementary table 1)[29–32]. The EMIm cation is unique among other cations since it provides delocalized positive charge around the imidazolium ring, effectively increasing the cation-anion distance and affording lower viscosities than ILs with other cations, owing to reduced Coulomb (electrostatic) interactions between ion pairs[53]. Second, the SEI components are unique with the inclusion of AlO$_x$ and NaCl due to Na reaction/passivation by chloroaluminate species, which facilitates the stabilization of Na plating/stripping cycling. This led to a cycle life of over 700 cycles, the longest among all the reported IL-based Na cells[29–31,41] (Fig. 4d).

We found that although FSI anions was indispensable for a stable SEI in our system, FSI alone was not sufficient for long cycle life of Na negative electrode. This was based on inferior cycling stability of Na/NVP@rGO cell in a non-chloroaluminate based electrolyte 1 M NaFSI in [EMIm]FSI IL electrolyte, displaying low and fluctuating CEs of only ~90%, despite the fact that it had a much higher FSI anion concentration of ~6 M compared with only ~0.2 M in the buffered Na–Cl–IL electrolyte (Supplementary Fig. 21). Similarly, the Na/NVP@rGO cell using NaFSI in N-propyl-N-methylpyrrolidinium bis(fluorosufonyl) imide IL electrolyte (molar ratio of 2:8) showed fluctuating CEs after ~ 65 cycles when cycling at 150 mA g$^{-1}$ (Supplementary Fig. 22). In addition, an inferior rate performance was demonstrated using NaFSI/N-propyl-N-methylpyrrolidinium bis(fluorosufonyl)imide IL electrolyte compared with that based on buffered Na–Cl–IL electrolyte (Supplementary Fig. 23).

Another important aspect was that previous IL electrolytes with highly concentrated F-based species (e.g., over 5 M of FSI anion concentration in NaFSI-[N-propyl-N-methylpyrrolidinium]FSI electrolyte with a molar ratio of 2:8)[32] were much higher in cost than conventional organic electrolytes due to expensive FSI species. A

much lower FSI concentration of only ~0.2 M was needed for the buffered Na–Cl–IL electrolyte, and at the same time reaching better cell performances (power density, CE, cycle life and discharge voltage etc.) than previous room temperature IL electrolytes (Supplementary Table 1). The buffered Na–Cl–IL electrolyte could be a promising candidate for affordable, high-safety energy storage towards real-world applications.

In conclusion, we develop a non-flammable and highly conductive ionic liquid electrolyte for high-energy/high-voltage Na metal batteries. The ionic liquid electrolyte is comprised of $AlCl_3$, NaCl and [EMIm]Cl and allows reversible Na plating/stripping upon addition of two additives, i.e., ethylaluminum dichloride and 1-ethyl-3-methylimidazolium bis(fluorosulfonyl)imide. The Na metal cells with NVP and NVPF positive electrodes achieve high CE up to 99.9%, and high energy and power density of 420 Wh $kg^{-1}$ and 1766 W $kg^{-1}$, respectively. Over 90 % of the original capacity can be retained after over 700 galvanostatic charge-discharge cycles. The solid-electrolyte interphase (SEI) probed by XPS and Cryo-TEM shows that the major components included NaCl, $Al_2O_3$ and NaF. The non-flammable and highly conductive IL electrolyte can serve as a promising candidate for sodium batteries with high safety and high performance, and can be potentially extended to a broad range of rechargeable battery systems such as Li and K batteries.

## Methods

**Preparation of IL electrolytes**. IL electrolytes were prepared in an Ar-filled glove box with water and oxygen content below 2 ppm. $[EMIm]Al_xCl_y$ IL was first made by mixing 1-ethyl-3-methylimidazolium chloride ([EMIm]Cl) and anhydrous $AlCl_3$ ( ≥ 99.0%, Fluka). [EMIm]Cl was dried at 80 ºC under vacuum for 24 h to remove residual water. For a certain molar ratio, e.g., 1.5 of $AlCl_3$/[EMIm]Cl, 1.78 g of [EMIm]Cl and 2.4 g of $AlCl_3$ were weighed in two glass vials, respectively. A small portion of $AlCl_3$ was then slowly added into [EMIm]Cl to avoid dramatic heat generation during the mixing. This process was repeated until all the $AlCl_3$ were introduced, and the mixture was stirred until all the solid was dissolved, followed by adding around 0.3 g of aluminum foil for purification. 1.8 g of the obtained light-yellow, clear liquid was kept at 70 ºC for 1 h under vacuum for removal of water, followed by adding 0.172 g NaCl (99.999 %, Sigma-Aldrich) and allowed to stir for 24 h. The supernatant was collected, and stirred with 1 wt% $EtAlCl_2$ (Sigma-Aldrich) for 1 h. The mixture was further added with 4 wt% [EMIm]FSI (dried at 70 ºC under vacuum for 12 h before use) and allowed to stir for 6 h to obtain the buffered + $EtAlCl_2$/[EMIm]FSI additive electrolyte. To avoid water absorption of the prepared IL electrolyte, all the agents were stored inside tightly closed and sealed bottles, and placed in Ar-filled glove box. [EMIm]Cl and NaCl were dried via heating under vacuum before use. [EMIm]FSI and N-propyl-N-methylpyrrolidinium bis(fluorosufonyl)imide were dried under vacuum at 70 ºC for 12 h before dissolving NaFSI salt. 1 M $NaClO_4$ in EC/DEC (1:1 by vol) with 5 wt % FEC was prepared as conventional organic electrolyte for comparison.

**Preparation of NVP@rGO and NVPF@rGO**. Graphene oxide (GO) was synthesized via a modified Hummer's method with more details described in Supplementary Information[54]. To prepare NVP@rGO, 0.69 g of $NH_4H_2PO_4$, 0.318 g of $Na_2CO_3$ and 0.364 g of $V_2O_5$ were dispersed in deionized water, followed by adding 0.72 g of oxalic acid ( ≥ 99.0 %, Sigma-Aldrich) at 70 ºC. The mixture was added with 7.3 mL GO aqueous dispersion (11 mg $mL^{-1}$) under vigorous stirring, and then freeze-dried to obtain the solid NVP@GO precursor. The precursor was grounded using an agate mortar, followed by sintering at 850 ºC with a heating rate of 2 ºC $min^{-1}$ in Ar to obtain the NVP@rGO powder. NVPF@rGO was prepared via a one-step hydrothermal method. Briefly, 0.536 g of NaF, 3.51 g of $NaH_2PO_4·2H_2O$ and 1.763 g of $VOSO_4·xH_2O$ (degree of hydration 3–5, Sigma-Aldrich) were dissolved in 30 mL deionized water, followed by mixing with 7.8 mL of GO aqueous dispersion (11 mg $mL^{-1}$) for 1 h to obtain a uniform dispersion. The mixture was immediately transferred into a 45 mL Teflon-lined stainless steel autoclave and kept at 120 ºC for 10 h. The resulted precipitates were centrifuged at 4,000 rpm using deionized water for 5 times, and the obtained solid was dried at 80 ℃ for 10 h in a vacuum oven to obtain the NVPF@rGO powder. For bare NVPF, no GO was added with all the other procedures remained the same.

**Electrochemical measurements**. All the electrochemical measurements were conducted at room temperature (22 ℃) unless otherwise specified. To prepare slurries, 70 wt% NVP@rGO or NVPF@rGO powder was mixed with 20 wt% conductive carbon black (Super C65, TIMICAL) and 10 wt% polyvinylidene fluoride (PVDF, $M_w$ = 180,000, Sigma-Aldrich) in N-methyl-2-pyrrolidone (NMP, 99.5 %, Sigma-Aldrich). The mixture was stirred for 10 h until a uniform and viscose slurry was obtained, which

was coated on a Mitsubishi carbon fibre paper (M30 type, 30 g $m^{-2}$). The electrodes were baked in a 120 ℃ vacuum oven for 2 h for removal of the residual NMP. The electrochemical performances were measured in pouch-type cells. Briefly, carbon tap (Ted Pella) was used to paste the positive electrode (Cu or Pt foil, NVP@rGO or NVPF@rGO electrodes) and negative electrode of Na metal foil onto an aluminum laminated pouch. The Na foil was prepared by rinsing a Na cube (99.9 %, Sigma-Aldrich) in anhydrous dimethyl carbonate ( ≥ 99.0 %, Sigma-Aldrich) for removal of the mineral oil on surface, cutting off the surface oxidation with blades, and pressing a fresh piece into a thin foil. Two nickel tabs (EQ-PLiB-NTA3, MTI) and a piece of glass fibre filter paper (GF/A, Whanman) were served as the current collector and separator, respectively. The obtained pouch was heated in an 80 ℃ vacuum oven for 8 h, and then transferred into an argon-filled glove box with water and oxygen content below 2 ppm to fill in the electrolyte (200 uL for each cell). The pouch was heat-sealed in the glove box before transferring out for further electrochemical measurement. Cyclic voltammetry was performed on a CHI760E electrochemical work station. The charge-discharge performances of the cells were measured with a Neware battery testing system (CT-4008-5V50mA-164-U). All the cells were allowed to age for 6 h before charge-discharge measurement. The specific capacity, energy and power density were calculated based on the total mass of NVP@rGO and NVPF@rGO.

**Characterization**. For Raman spectra, IL electrolytes were injected and sealed into transparent plastic pouches in an Ar-filled glove box. The spectra were acquired (250–500 $cm^{-1}$) using an $Ar^+$ laser (532 nm) with 0.8 $cm^{-1}$ resolution. The conductivity measurement was performed on a conductivity meter (FiveEasy Plus, Mettler Toledo). Prior to characterization, the electrodes were rinsed with anhydrous dimethyl carbonate for 6 times, and dried under vacuum at room temperature. They were further sealed in Ar-filled pouches and quickly transferred into the vacuum chamber to avoid too much exposure to air. The Na ion concentration of the buffered Na–Cl–IL electrolyte was measured using a Thermo Scientific ICAP 6300 Duo View Spectrometer. SEM images were acquired from a Hitachi/S-4800 SEM operated at 15 kV, and EDS analysis was performed on a Horiba/Ex-450 EDS spectroscopy. FIB-SEM was performed on a dual-beam field-emitting focused ion beam microscope (VERSA 3D DualBeam) with an accelerating voltage of 20 kV. TEM image of NVP@rGO was obtained with a JEOL JEM-2100F operated at 200 kV. XRD pattern was measured with a Bruker D8 Advance powder X-ray diffractometer with Cu Kα radiation. TGA measurement was performed on a PerkinElmer/Diamond TG/DTA thermal analyser at a heating rate of 5 ℃ $min^{-1}$ in air for NVP@rGO and NVPF@rGO, and in nitrogen for IL and organic electrolyte, respectively. The temperature range used for determining rGO percentage was 180–460 ℃, and the weight loss below 180 ℃ was due to water removal that is also used to determine the water content of products synthesized in aqueous solution[55]. XPS spectra were collected on a PHI 5000 VersaProbe Scanning XPS Microprobe. All the binding energy values were calibrated with C1s peak (284.6 eV). Depth profile was conducted using Ar ion sputtering at 1 kV and 0.5 μA over a 2 × 2 mm area, corresponding to a $SiO_2$ sputter rate of 2 nm $min^{-1}$. Glass fibre separators soaked with electrolyte were used to test the flammability of the electrolyte. Cryo-TEM was performed on an FEI Titan Krios cryogenic transmission electron microscope operated at 300 kV. Na was plated on a Cu TEM grid in a 2032 type coin cell at a current density of ~ 0.2 mA $cm^{-2}$ for 30 min, using 150 uL Na–Cl–IL and one glass fibre as electrolyte and separator, respectively. The coin cell was disassembled in an Ar-filled glove box, followed by removing the residual electrolyte on Na-plated Cu TEM grid using anhydrous DMC and drying it under vacuum. The TEM grid was then carefully mounted onto a TEM cryo-holder and transferred into the chamber of Cryo-TEM without exposing to air. Similar processes were performed for element mapping using a FEI Titan Themis 60-300 transmission electron microscope equipped with a cooling sample holder.

## Data availability

The data that support the plots within this paper and other findings of this study are available from the corresponding author upon reasonable request.

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

## Acknowledgements

Part of this work was supported by the Stanford Bits and Watts Program and gift funds. Part of this work was performed at the Stanford Nano Shared Facilities (SNSF), supported by the National Science Foundation under award ECCS-1542152.

## Author contributions

H.S. and H.D. conceived the idea for the project. H.S., X.X., Y.M. and Y.K. prepared Na₃V₂(PO₄)₃@reduced graphene oxide and Na₃V₂(PO₄)₂F₃@reduced graphene oxide. H.S., and X.X. performed electrochemical experiments. H.S., G.Z. and M.A. conducted Raman spectroscopy measurements. G.Z. and H.S. performed X-ray photoelectron spectroscopy measurements. H.S., Y.-Y.L., W.H. and M.L. performed and analyzed focused ion beam, scanning electron microscope, energy-dispersive X-ray spectroscopy, thermogravimetric analysis and X-ray diffraction measurements. J.L. and H.S. performed the inductively coupled plasma measurement. M.G., Y.Z. and H.S. performed cryogenic transmission electron microscope and scanning transmission electron microscopy measurements. M.-C.L. performed the ionic conductivity measurements. H.D. supervised the project. H.S., H.P. and H.D. prepared the manuscript. All authors participated in experimental data analysis and result discussion.

## Additional information

**Competing interests:** The authors declare no competing interests.

