## [Peer Review File · Nature Communications]

Reviewers' comments:

Reviewer #1 (Remarks to the Author):

This manuscript entitled "A Safe, Non-Flammable Sodium Metal Battery Based on a Novel Ionic Liquid Electrolyte" presented a novel electrolyte based on chloroaluminate ionic liquid for safe sodium metal battery. The proposed ionic liquid electrolyte was fire-resistant and had a high ionic conductivity. Furthermore, the additives were demonstrated to be crucial for the reversible sodium plating and stability of NVPF hybrid electrode. The concept is new, and the results are satisfying. Therefore, I would like to recommend it to be published after some minor revisions and addressing some issues below.

1. The chloroaluminate ionic liquids have high tendency to adsorb water. What method does the author use to keep them water-free, besides heating to dry? If possible, a water content report is preferred for this matter.
2. One of the problems of using ionic liquid is that the effective transference number of Na⁺ may not be too high, even though they possess high ionic conductivity, especially Al-based salt is also present. The author should give a relative detection of the sodium ion transference number.
3. It is understandable that the [EMIm]FSI has been demonstrated to be critical role of Na plating/stripping, and the sole [EMIm]FSI is not suitable for Na/NVP@rGO cell. Is it possible using [EMIm]FSI as solvent and EtAlCl₂ as additive for the cell?
4. The author claim that Na⁺ is the main intercalation ion when applied in NVP and NVPF cells. How to eliminate the influence of EMIm⁺ and AlCl₄⁻ as the intercalation ions as well?
5. In the investigation of the sodium anode by mapping, Al is also found to be deposited on the Na. Does that affect the actual reference electrode voltage?
6. Some recent published works are recommended to be cited for a comparison. These include: (i) Nature Communications 9 (2018) 3870; (ii) Angew. Chem. Int. Ed. 57 (2018) 10168; (iii) ACS Nano 12 (2018) 12337.

Reviewer #2 (Remarks to the Author):

General comments

Most of the work on sodium-ion batteries involves the use of organic solutions as electrolytes, in spite of their safety problems. It is important to find suitable safe electrolytes for these devices. In this regard, the work shows interesting results and a thorough investigation of the Na-Cl-IL electrolyte when used in a Na-ion battery, specially the composition of the SEI layer.

However, AlCl₄⁻-[EMIm]⁺ ionic liquid has been studied for a long time, and its properties - such as high stability, non-flammability and high conductivity - are well established. Ionic liquids high stability in comparison to organic solvents, in particular, is the main reason why ILs have been studied as Li-ion and Na-ion battery electrolytes.

The good results found for the NVPF@rGO|Na cell are not necessarily due to the high conductivity of the ionic liquid used as electrolyte. It is desirable to make a comparison among different ionic liquids using the same cell assembly and the same positive electrode material. One suggestion is the N-

propyl-N-methylpyrrolidinium bis(fluorosulfonyl)imide, or C3mpyrFSI. This type of comparison would be more insightful than showing results for two similar positive electrodes (NVP@rGO and NVPF@rGO).

Specific comments

For secondary (rechargeable) batteries, it is recommended to use the terms 'negative' and 'positive' for the cell electrodes, instead of 'anode' and 'cathode.'

The procedure for preparing graphene oxide is mentioned as a modified Hummer's method (line 373). What are the details of the method actually employed?

Different cell assemblies were used, with different materials for the positive electrode. How was Na foil used as positive electrode (line 398)?

1-ethyl-3-methylimidazolium chloride name needs to be reviewed in line 60.

Supplementary figure 14 was described as TGA profile for NVPF@rGO in the text (line 221), but subtitled as TGA profile for NVP@rGO. In any case, it is unclear how the rGO percentage was determined in the sample, since there is continuous weight loss until ~450 °C. It would be interesting to specify the temperature range used.

Reviewer #1:

Remarks to the Author: *This manuscript entitled “A Safe, Non-Flammable Sodium Metal Battery Based on a Novel Ionic Liquid Electrolyte” presented a novel electrolyte based on chloroaluminate ionic liquid for safe sodium metal battery. The proposed ionic liquid electrolyte was fire-resistant and had a high ionic conductivity. Furthermore, the additives were demonstrated to be crucial for the reversible sodium plating and stability of NVPF hybrid electrode. The concept is new, and the results are satisfying. Therefore, I would like to recommend it to be published after some minor revisions and addressing some issues below.*

Response: We thank the reviewer’s positive/insightful comments. The comments and suggestions have helped a lot to further strengthen this work. The manuscript has been carefully revised according to your valuable suggestions.

(1) The chloroaluminate ionic liquids have high tendency to adsorb water. What method does the author use to keep them water-free, besides heating to dry? If possible, a water content report is preferred for this matter.

Response: Thank you very much for your comments and advices. Water content report was not provided because when injecting the chloroaluminate ionic liquids into a Karl Fischer titrator, precipitation was observed in the testing electrolyte containing iodine-based species, probably due to the reaction with AlCl_3 -based species. Actually, any water in the Na-Cl-IL will react with aluminum chloride-based species and become HCl, which was supposed to be removed by heating under vacuum and reacting with EtAlCl_2 .

Revisions made: We added more details to ensure the ionic liquid electrolyte water-free which was described at the last paragraph of Page 11, as follows:

“To avoid water absorption of the prepared IL electrolyte, all the agents were stored inside tightly closed and sealed bottles, and placed in Ar-filled glove box. $[\text{EMIm}]\text{Cl}$ and NaCl were dried via heating under vacuum before use.”

(2) One of the problems of using ionic liquid is that the effective transference number of Na^+ may not be too high, even though they possess high ionic conductivity, especially Al-based salt is also present. The author should give a relative detection of the sodium ion transference number.

Response: Thank you very much for the insightful suggestion. The Bruce-Vincent method that is generally used might need some modification to accurately reveal the sodium ion transference number of the Na-Cl-IL electrolyte, particularly with the presence of Al-based salt as you mentioned, which makes the system more complicated than conventional organic and IL electrolyte systems. As a result, we have not included the sodium ion transference number herein, but will design more

experiments to investigate this system in the future. Thanks again for your comments and advices!

(3) It is understandable that the [EMIm]FSI has been demonstrated to be critical role of Na plating/stripping, and the sole [EMIm]FSI is not suitable for Na/NVP@rGO cell. Is it possible using [EMIm]FSI as solvent and EtAlCl₂ as additive for the cell?

Response: We performed additional experiments to address this issue. Using [EMIm]FSI as solvent (e.g., 50 wt% in the Na-Cl-IL electrolyte) made the obtained electrolyte highly viscous, and the Na/NVPF@rGO battery showed large overpotential and low specific capacity as below.

Figure R1. Galvanostatic charge-discharge curves of a Na/NVP@rGO cell using Na-Cl-IL electrolyte with 50 wt% [EMIm]FSI.

In addition, we also made a Na/NVP@rGO cell using 1 M NaFSI in [EMIm]FSI (as solvent) IL electrolyte. It showed low and fluctuating CEs as below, indicating less stable SEI formed for reversible Na plating and stripping than our buffered IL electrolyte.

Figure R2. Capacity and Coulombic efficiency retention of a Na/NVP@rGO cell using 1 M NaFSI in [EMIm]FSI IL electrolyte at 150 mA/g.

Please note that another factor/consideration is that due to the high price of [EMIm]FSI, it is preferable to reduce its amount in forming electrolytes to achieve high performance and low cost at the same time.

(4) The author claim that Na⁺ is the main intercalation ion when applied in NVP and NVPF cells. How to eliminate the influence of EMIm⁺ and AlCl₄⁻ as the intercalation ions as well?

Response: Thank you for your insightful comments. AlCl₄⁻ and EmIm⁺ demonstrate much larger radii of ~ 2.8 Å and 2×2.7 Å (non-spherical), respectively, compared with ~1.2 Å of Na⁺ (Wasserscheid, P. & Welton, T. Ionic liquids in Synthesis, John Wiley & Sons, 2008). The intercalation voltage for AlCl₄⁻ and EMIm⁺ into NVP and NVPF (if possible) should be higher than Na⁺ intercalation voltage. However, no extra voltage plateaus were observed except normal Na intercalation plateaus (3.45 V vs. Na for NVP@rGO and 3.75 V/4.2 V vs. Na for NVPF@rGO) within the applied voltage ranges. As a result, we believe Na⁺ is the main intercalation ion in this system.

(5) In the investigation of the sodium anode by mapping, Al is also found to be deposited on the Na. Does that affect the actual reference electrode voltage?

Response: Thank you for the question. Because the atomic fraction of Al was only ~ 3.4 % in the EDS mapping result, it should have very little influence on the voltage of the Na electrode. Indeed, the full cell voltage (e.g., ~ 3.3 V for Na/NVP battery discharging) matches the voltage relative to the ref. electrode (positive electrode: ~ 3.3 V vs. Na, negative electrode: 0 V vs. Na).

(6) Some recent published works are recommended to be cited for a comparison. These include: (i) Nature Communications 9 (2018) 3870; (ii) Angew. Chem. Int. Ed. 57 (2018) 10168; (iii) ACS Nano 12 (2018) 12337.

Response: Thank you for the suggestions. We have referred the mentioned works as new Refs. 4-6.

Reviewer #2:

Remarks to the Author: *Most of the work on sodium-ion batteries involves the use of organic solutions as electrolytes, in spite of their safety problems. It is important to find suitable safe electrolytes for these devices. In this regard, the work shows interesting results and a thorough investigation of the Na-Cl-IL electrolyte when used in a Na-ion battery, specially the composition of the SEI layer.*

Response: We thank the reviewer's positive/insightful comments. By addressing the issues raised, we have strengthened this work substantially. We have also investigated the cyclic stability and rate performances of Na/NVP@rGO cells using NaFSI/N-propyl-N-methylpyrrolidinium bis(fluorosulfonyl)imide IL electrolyte as a comparison according to your valuable suggestions.

(1) AlCl₄-[EMIm]⁺ ionic liquid has been studied for a long time, and its properties - such as high stability, non-flammability and high conductivity - are well established. Ionic liquids high stability in comparison to organic solvents, in particular, is the main reason why ILs have been studied as Li-ion and Na-ion battery electrolytes. The good results found for the NVPF@rGO/Na cell are not necessarily due to the high conductivity of the ionic liquid used as electrolyte. It is desirable to make a comparison among different ionic liquids using the same cell assembly and the same positive electrode material. One suggestion is the N-propyl-N-methylpyrrolidinium bis(fluorosulfonyl)imide, or C3mpyrFSI. This type of comparison would be more insightful than showing results for two similar positive electrodes (NVP@rGO and NVPF@rGO).

Response: Thank you very much for your constructive suggestions. We performed additional experiments to address this issue. Na/NVP@rGO cells were made using NaFSI/N-propyl-N-methylpyrrolidinium bis(fluorosulfonyl)imide (molar ratio: 2:8) IL electrolyte for comparison of both rate capability and cyclic stability.

For cyclic stability of NaFSI/N-propyl-N-methylpyrrolidinium bis(fluorosulfonyl)imide (molar ratio: 2:8) electrolyte-based battery, the Coulombic efficiency started to fluctuate after ~ 65 cycles (Figure R1 below), indicating dendrite formation due to a lack of stable SEI on Na negative electrode.

Figure R1. Capacity and Coulombic efficiency retention of a Na/NVP@rGO cell using NaFSI/N-propyl-N-methylpyrrolidinium bis(fluorosulfonyl)imide IL electrolyte (molar ratio: 2:8). Current density, 150 mA/g.

For rate performance, Na-Cl-IL electrolyte-based battery demonstrated a better rate performance at high current densities compared with NaFSI/N-propyl-N-methylpyrrolidinium bis(fluorosulfonyl)imide electrolyte (Figure R2).

Figure R2. Comparison of galvanostatic charge-discharge curves of Na/NVP@rGO cells using (a) Na-Cl-IL and (b) NaFSI/N-propyl-N-methylpyrrolidinium bis(fluorosulfonyl)imide (molar ratio: 2:8) electrolytes at varied current densities from 25 to 400 mA/g.

Also, a Na/NVP@rGO cell using 1 M NaFSI in [EMIm]FSI as IL electrolyte showed low and fluctuating CEs as shown in Figure R3.

Figure R3. Capacity and Coulombic efficiency retention of a Na/NVP@rGO cell using 1 M NaFSI in [EMIm]FSI IL electrolyte at 150 mA/g.

Revisions made: We added the comparisons of cyclic stability and rate performance based on NaFSI/N-propyl-N-methylpyrrolidinium bis(fluorosulfonyl)imide (molar ratio: 2:8) IL electrolyte at the third paragraph of Page 10 as follows:

“Similarly, the Na/NVP@rGO cell using NaFSI in N-propyl-N-methylpyrrolidinium bis(fluorosulfonyl)imide IL electrolyte (molar ratio of 2:8) showed fluctuating CEs after ~ 65 cycles when cycling at 150 mA/g (Supplementary Fig. 22). In addition, an inferior rate performance was demonstrated on the basis of NaFSI/N-propyl-N-methylpyrrolidinium bis(fluorosulfonyl)imide IL electrolyte compared with that using buffered Na-Cl-IL electrolyte (Supplementary Fig. 23).”

We also added two new Supplementary Figs. 22 and 23 as follows:

Supplementary Fig. 22. Galvanostatic charge-discharge curves of a Na/NVP@rGO cell using NaFSI/N-propyl-N-methylpyrrolidinium bis(fluorosulfonyl)imide (molar ratio: 2:8) IL electrolyte at varied current densities from 25 to 400 mA/g.

Supplementary Fig. 23. Capacity and Coulombic efficiency retention of a Na/NVP@rGO cell using NaFSI/N-propyl-N-methylpyrrolidinium bis(fluorosulfonyl)imide IL electrolyte (molar ratio: 2:8). Current density, 150 mA/g.

(2) For secondary (rechargeable) batteries, it is recommended to use the terms ‘negative’ and ‘positive’ for the cell electrodes, instead of ‘anode’ and ‘cathode.’

Revisions made: ‘Anode’ and ‘cathode’ have been changed to ‘negative electrode’ and ‘positive electrode’ as suggested.

(3) The procedure for preparing graphene oxide is mentioned as a modified Hummer’s method (line 373). What are the details of the method actually employed?

Revisions made: The preparation details of graphene oxide were added in Supplementary information as follows:

“Preparation of graphene oxide. 1 g flake graphite powder was pre-oxidized in the mixture of 30 mL sulfuric acid and 10 mL nitric acid under stirring for 24 h. After washing with deionized water and drying, the obtained powder was exfoliated in a tube furnace at 1000 °C for 10 s, followed by reacting with 60 mL oleum, 0.84 g $K_2S_2O_8$ and 1.3 g P_2O_5 at 80 °C for 5 h under stirring. After cooling down to room temperature, 500 mL deionized water was slowly added to the suspension, and the dried products were obtained by vacuum filtrating and washing for 3 times, and dried in a vacuum oven. The resulted powder was added to 50 mL oleum in ice bath, followed by adding 3 g $KMnO_4$ slowly under vigorous stirring, during which the temperature was kept below 20 °C. The mixture was then heated to 35 °C and stirred for another 2 h, and diluted with 500 mL deionized water and added with 2 mL 30 wt% H_2O_2 . The dispersion was left overnight, and the brown gel at bottom was washed with deionized water, followed by centrifuging with 1 M HCl solution for 5 times, and then washing with deionized water until the decantate turned nearly neutral.”

(4) Different cell assemblies were used, with different materials for the positive electrode. How was Na foil used as positive electrode (line 398)?

Revisions made: As suggested, we have revised it at the last paragraph of Page 12 as follows:

“Briefly, carbon tap (Ted Pella) was used to paste the positive electrode (Cu or Pt foil, NVP@rGO or NVPF@rGO electrodes) and negative electrode of Na metal foil onto an aluminum laminated pouch.”

(5) 1-ethyl-3-methylimidazolium chloride name needs to be reviewed in line 60.

Revisions made: We have revised it at the second paragraph of Page 2 as follows:

“Among them, ILs comprised of AlCl₃ and 1-ethyl-3-methylimidazolium chloride ([EMIm]Cl) are a classical chloroaluminate based electrolyte system with many desired properties...”

(6) Supplementary figure 14 was described as TGA profile for NVPF@rGO in the text (line 221), but subtitled as TGA profile for NVP@rGO. In any case, it is unclear how the rGO percentage was determined in the sample, since there is continuous weight loss until ~450 °C. It would be interesting to specify the temperature range used.

Response: The temperature range used for determining rGO percentage is 180-460 °C. The weight loss below 180 °C was due to the water removal of NVPF@rGO, which is generally used for determination of the water content of products synthesized in aqueous solution (*ACS Energy Lett.* 2017, **2**, 1122-1127).

Revisions made: We have specified temperature range at the last paragraph of Page 13 with a new Ref. 55 as follows:

“The temperature range used for determining rGO percentage was 180-460 °C, and the weight loss below 180 °C was due to water removal that is also used to determine the water content of products synthesized in aqueous solution⁵⁵.”

REVIEWERS' COMMENTS:

Reviewer #1 (Remarks to the Author):

The authors have addressed all my comments. I recommend this revised manuscript to be published in Nature Communications.

Reviewer #2 (Remarks to the Author):

All my concerns regarding the first version of the article were answered by the authors; so that, in my opinion, the manuscript is now appropriate to be published in Nature Communications.

Reviewer #1:

Remarks to the Author: The authors have addressed all my comments. I recommend this revised manuscript to be published in Nature Communications.

Response: Thanks a lot for your review and suggestions on our manuscript.

Reviewer #2:

Remarks to the Author: All my concerns regarding the first version of the article were answered by the authors; so that, in my opinion, the manuscript is now appropriate to be published in Nature Communications.

Response: We appreciate the reviewer's important suggestions and comments to help us enhance this work. Thanks a lot!